# Prevalence of Oral and Maxillofacial Disorders in Patients with Systemic Scleroderma—A Systematic Review

**DOI:** 10.3390/ijerph18105238

**Published:** 2021-05-14

**Authors:** Korbinian Benz, Christine Baulig, Stephanie Knippschild, Frank Peter Strietzel, Nicolas Hunzelmann, Jochen Jackowski

**Affiliations:** 1Department of Oral Surgery and Policlinical Ambulance, Faculty of Health, Witten/Herdecke University, Alfred-Herrhausen-Str. 45, 58448 Witten, Germany; jochen.jackowski@uni-wh.de; 2Faculty of Health, Institute for Medical Biometry and Epidemiology, Witten/Herdecke University, Alfred-Herrhausen-Str. 50, 58448 Witten, Germany; christine.baulig@uni-wh.de (C.B.); stephanie.knippschild@uni-wh.de (S.K.); 3Charité Centre for Dentistry, Department Periodontology, Oral Medicine and Oral Surgery, Charité-University Berlin, Assmannshauser Str. 4-6, 14197 Berlin, Germany; frank.strietzel@charite.de; 4Department of Dermatology and Venerology, University of Cologne, Kerpener Str. 62, 50937 Köln, Germany; nico.hunzelmann@uni-koeln.de

**Keywords:** systemic scleroderma, orofacial manifestations, mucosal changes, skin fibrosis, rare disease, pooled effect estimates

## Abstract

Background: Systematic scleroderma is a rare chronic autoimmune disease of unknown aetiology. The aim of this study was to identify the prevalence of orofacial pathognomonic conditions in patients with systemic scleroderma using only randomised prospective studies that investigated the treatment of oral and maxillofacial changes, highlighted associations between the disease and Sjogren’s syndrome, and/or analysed the effect of oral hygiene. Methods: The literature was systematically reviewed based on Cochrane Library, EMBASE, PubMed, Scopus, and Web of Science articles published up to March 2020. The primary endpoint of this analysis was defined as an estimation of the prevalence of oral mucosal changes in different areas of the oral cavity (oral mucosa, tongue, lip, periodontal status, bones, and other regions) in patients suffering from scleroderma. Therefore, a systematic literature search (Cochrane Library, EMBASE, PubMed, Scopus, and Web of Science) was conducted and limited by the publication date (1950-03/2020) and the publication language (English). Extracted frequencies were pooled using methods for meta-analysis. In order to obtain the highest level of evidence, only prospective study reports were considered to be eligible. Results: After full-text screening, 14 (766 patients) out of 193 publications were eligible for the final analysis. Twelve studies produced reliable results in the final data sets. Calculation of the pooled effect estimate (random effects model) revealed a prevalence of 57.6% (95% CI: 40.8–72.9%) for the main area “lip”. For the area “oral mucosa”, a prevalence of 35.5% (95% CI: 15.7–62.0%) was calculated. The prevalence for “other regions” was only based on studies with salivary changes and was calculated to be 25.4% (95% CI: 14.2–41.3%). Conclusion: The most pathognomonic conditions in the orofacial region in patients with systemic scleroderma affect the lips, oral mucosa, and salivary glands.

## 1. Introduction

Systemic scleroderma (SSc) is defined as being part of the complex of inflammatory fibrotic diseases called collagenoses. It is a chronic inflammatory disease of the vascularised connective tissue with circumscribed or generalised fibrosis of the skin and inner organs and is characterised by an excessive incorporation of collagen, activation of the immune system, vascular hyperreactivity, and obliterating microvascular phenomena. Sufferers of systemic scleroderma show antinuclear antigens (ANAs) in more than 90% of all cases, with anti-Scl-70 (approximately 30%) and anti-centromere antibodies being the most abundant [1,2].

The prevalence of systemic scleroderma is estimated to vary between 4 and 40 cases in every 100,000 persons [3], mainly affecting women at a ratio of 3:1 or more vs. male patients [4,5]. Major differences might exist region to region. According to estimates, Northern Europe and Japan have a lower prevalence (<150 per million) and an incidence of fewer than 10 per million per year, whereas a prevalence of 276–443 per million and an incidence of 14–21 per million per year are assumed for Southern Europe, Australia, and North America [6]. As a general rule, the disease becomes manifest between 40 and 50 years of age [1].

Systemic sclerosis belongs to the group of systemic autoimmune diseases with an aethiopathogenesis that cannot or can only incompletely be explained at the present juncture. Disorders of the physiological regulatory circuit of the immune system may result in dysfunctions of various organ systems as a reaction. Accordingly, autoimmune disease is a possible differential diagnosis that must be taken into account in patients with multiple symptoms with unclear interrelationships. This multiplicity of symptoms and disparate clinical courses result from the different functions of the affected organs.

The American College of Rheumatology (ACR), in cooperation with the European League Against Rheumatism (EULAR), has devised a classification system aimed at delimitating SSc against other connective tissue diseases [7]. Systemic scleroderma is divided into two subtypes, which differ by the degree of skin involvement. In limited cutaneous systemic scleroderma (IcSSc), the extremities are affected down to the elbow and knee joints at maximum, and the face (nose and/or chin) also exhibits symptoms. In this type, organ involvement is less common and, in most cases, less severe. An IcSSc subtype is the CREST syndrome (calcinosis cutis, Raynaud’s syndrome, oesophageal symptoms, sclerodactyly, and telangiectasia). In diffuse cutaneous systemic scleroderma (dcSSC), the sclerosis may affect the entire body. Within a year, it can lead to severe fibrosis, with inner organs also commonly affected [8,9]. It is diagnosed and classified by antibody testing. IcSSc is more often associated with centromere-specific antibodies, whereas topoisomerase I- or RNA polymerase III-specific antibodies are more frequently found in dcSSc [10]. Patients with the limited type generally have a better prognosis, although 10–15% of the patients will eventually develop pulmonary-arterial hypertension, approximately 80% gastrointestinal involvement, and/or approximately 30–40% an interstitial pulmonary disease [11]. An involvement of the viscera is also responsible for a reduced life expectancy in patients with diffuse types of scleroderma. Tendon, vascular, or joint damage, which may develop in some cases, hinders sufferers in their life management and thus impacts their quality of life [12]. Scleroderma’s multifaceted nature as a disease that involves circulatory disturbances, hardening of the connective tissue, inflammation, and immune system dysfunctions always requires interdisciplinary care of the patient. Depending on the involved organ systems, a wide range of specialties are involved, including dermatology, rheumatology, gastroenterology, cardiology, pneumology, nephrology, and oral medicine [13]. The involvement of the dermis entails a pathognomonic physiognomy that, not unlike Raynaud’s symptomatology or early development of a scleroglosson, may precede the actual diagnosis.

The face and mouth are affected in most SSc cases [14]. Organ-specific diagnostic testing shows that, out of 5500 patients registered in the German Network for Systemic Scleroderma, 24.1% reported the disease-associated co-involvement of the masticatory organ [15]. Typical symptoms are microstomy, microcheilia, perioral wrinkles, extraoral/intraoral telangiectasia (Figure 1a), scleroglosson, receding gums, and hyposalivation and/or dry mouth (Figure 1b).

This may also lead to periodontitis as a secondary manifestation of SSc [16,17]. Three main issues have been identified with the help of the mouth handicap in systemic sclerosis scale (MHISS) and generally play a critical role in SSc patients in the context of dental therapies [18]. On the one hand, patients suffer from a dry mouth, which can cause ulcerations and inflammation of the oral mucosa. On the other hand, a functional impairment of the masticatory organ and associated dysphagia will result. Together with the variably configured fibrosis, the retraction of the lips leading to the hallmark perioral wrinkles is responsible for impaired mouth opening (Figure 2).

Finally, reports have been published concerning treatment-specific adverse events that have been linked to dental therapies [19]. A clustering of oral cancer diseases has been observed in the diffuse type of SSc in particular, because the microstomy makes a dedicated intraoral examination difficult or even impossible in many cases [20,21,22].

About 8000 rare diseases are documented globally, of which about 15% are associated with changes in the masticatory organ [23]. SSc ranks among the rare diseases with orofacial involvement.

The aim of our study was to identify the prevalence of orofacial pathognomonic conditions in patients with systemic scleroderma using only randomised prospective studies that investigated the treatment of oral and maxillofacial changes, highlighted associations between the disease and Sjogren’s syndrome, and/or analysed the effect of oral hygiene.

## 2. Materials and Methods

In order to determine the prevalence of symptoms related to SSc, a focused question was formulated: Which symptoms of the oral and maxillofacial regions with which prevalence do patients with SSc suffer from? For a precise systematic literature search, the PICO-formate (Patients or Population: patients suffering from SSc; Intervention or exposure: which symptoms at the oral mucosa and perioral region do these patients suffer from?; Comparison: not applicable; Outcome: frequency of the symptoms in patients with SSc) was utilized to clearly define inclusion and exclusion criteria for publications.

### 2.1. Search Strategy

Therefore, existing literature pertaining to mouth and/or oral structures related to SSc was systematically searched on the basis of Cochrane Library, EMBASE, PubMed, Scopus, and Web of Science articles. K.B. and J.J. developed the search strategy. J.J. and K.B. applied the search. It was limited to English-language publications from March 1950 to March 2020. The study reports identified from the search engines were examined by reviewing the abstracts and full texts with regard to inclusion and exclusion criteria (see Table 1).

The search algorithm comprised the combinations of the terms in Table 2.

The main groups were developed from the search terms and could be further divided into subgroups (see Table 3).

### 2.2. Screening and Selection Process

The studies were selected following the PRISMA guidelines (see Figure 3).

Due to the low evidence by study design (RCT) in publications of the research area in scleroderma and in order to obtain the most valid information possible, only study reports from prospective study reports were included for the estimation of a pooled effect estimate (the prevalence of sclerodermy in special areas of the oral cavity). In addition, the studies had to comprise results from a minimum of ≥3 patients. Retrospective trials, reviews, meta-analyses, questionnaire surveys, health economic evaluations, animal experiments, and laboratory studies were excluded from final analysis. Furthermore, this investigation did not include case reports, comments, or letters to the editor. The prevalences were calculated from the full text information of all included studies by extracting the number of study participants with scleroderma and the number of patients with affected special oral areas. Raw data was documented using Excel^®^ (Office 2010 edition for Windows^®^, Microsoft, Redmond, Washington, DC, USA).

### 2.3. Data Analysis

Descriptive evaluation for the systematic review was carried out by absolute and relative frequencies using the software IBM SPSS Statistics 26 (IBM, New York, NY, USA). To estimate the pooled prevalence, Comprehensive Meta Analysis V2 (CMA^®^ software, edition 2.2.064, Biostat, Englewood, CO, USA) was used. Based on the reported number of included patients and patients with symptoms in special areas of the oral cavity (frequency per study), a random effects model assumption was made, and the DerSimonian-Laird estimator was used to estimate the T2 for each area. The pooled prevalence was then again presented by means of its 95% confidence interval and the underlying total number of patients in each study.

Statistical heterogeneity was explored by means of forest plots as well as I^2^ statistics (the iterative Paule-Mandel method to estimate between-study variance); an I^2^ value above 75% was considered to indicate substantial heterogeneity. Furthermore, funnel plots were used for graphical representations of dropout rates per trial (in logit scale) in relation to the respective trial size (in standard error scale) to account for asymmetric prevalence profiles among trial reports. In the case of significant heterogeneity among the dropout rates of the included trials, the Duval and Tweedie trim and fill method for publication bias was applied: the estimated prevalence was adjusted for the putative underreporting of trials, again by stressing the random effects model assumption, to derive a conservative adjusted pooled prevalence.

### 2.4. Assessment of Risk of Bias

The reporting quality was verified by determining and evaluating the following key items: the availability of a Consort flow diagram and a documented sample size calculation, whether it was a multicenter trial, whether a methodical department was part of the planning phase, and whether an ethical approval was documented. These items may not necessarily contain a secure reference to a correctly conducted clinical trial of high quality, but they reveal trial details to authors preparing their own publications in accordance with the relevant recommendations.

## 3. Results

The 2132 publications that were included in this analysis until March 2020 were initially located in the databases. After the duplicates (N = 485), articles in languages other than English (N = 64), and articles addressing a different disease or problem (N = 838) were discarded, 193 study reports remained. Two independent, full-text reviewers of these publications (K.B. and J.J.) eliminated 32 retrospective reports, 21 cross-sectional studies, 101 case reports/case series, 23 reviews, and 2 multiple publications (kappa = 0.81). Discrepancies were solved by discussion. The dataset thus obtained and therefore incorporated 12 reports that met the criteria for inclusion and were included in the analysis with a total of 766 patients during the period from 1983 to 2020. Study sizes spanned from 9 [24] to 218 participants [25] (see Table 4).

### 3.1. Results of Individual Studies

Three studies [26,27,30] with 235 patients looked at the interrelationship between SSc and Sjögren’s syndrome, which can be accompanied by xerostomia. Target criteria were the patients’ subjective sensations and histopathological lip assessments. Fibrosis and increased volume of the labial and parotid salivary glands were observed in 33% of cases, and Sjögren’s syndrome was found in 29% of cases [26].

Drosos et al. [27] reported a study of 44 scleroderma patients, diagnosing the comorbidity of Sjögren’s syndrome on the basis of lip biopsies. Ten patients (22%) showed high-grade focal lymphocyte infiltration, which was consistent with the diagnosis of Sjögren’s syndrome. Three patients (0.1%) showed low-grade infiltration. Seventeen patients (38%) were found to have “mild” to “moderately severe” fibrosis, whereas tissue structure was normal in 14 patients (31%). Histology was classified according to Tarpley et al. [36]. Moreover, an enlargement of the parotid gland was reported in 44.4% of cases (N = 20). Avouac et al. [30] determined a subjective sicca symptomatology in 68% (N = 91) of the participants by using a questionnaire. Following a positive Schirmer I test, which is based on the revised American-European Consensus Group for Systemic Sclerosis [37], 91 participants (68%) underwent lip biopsies. The histological examination revealed fibrosis of the studied tissue.

Fourteen patients (10%) met the criteria of a diagnosis of Sjögren’s syndrome. In a double-blind study, Naylor et al. [24] found a significant improvement of the mouth opening vs. a control group, an improvement that was affected by conventional mouth exercises (5.6 mm vs. 3.0 mm). The effect of domestic exercises was also documented by Pizzo et al. [29], who reported an improvement of the mouth opening in all microstomia subjects after 18 months (maximal mouth opening ≤30 mm) (mean increase: 10.7 ± 2.06 mm, *p* < 0.005). This effect was independent of whether the patients had dentition or not. In contrast, Rannou et al. [25] found no significant improvement through domestic physiotherapy based on the Health Assessment Questionnaire Disability Index (HAQ DI) in 218 patients assessed after 12 months. The literature reviewed in this work also provided descriptions of surgical procedures aimed at improving mouth opening, inter alia, by the injection of autologous fat tissue. According to data presented by Del Papa et al. [34], 80% (N = 16) of the patients were “very satisfied” and 20% (N = 4) were “satisfied” with the outcome. The maximal intercisal distance showed a statistically significant increase after three months (mean increase: 2.63 mm, paired t-value: 7.83, *p* < 0.0001).

Medication-based approaches in the treatment of the skin manifestations included applications of potassium aminobenzoate [28] and methotrexate [31]. In 1994, Clegg et al. investigated the effects of potassium aminobenzoate vs. a placebo. No significant improvement in terms of mouth opening, skin thickness, or lip mobility was noted [28], whereas the application of methotrexate in a study group of Krishna Sumanth et al. (2007) led to a significant improvement of the mouth opening (33.92 mm ± 1.70 mm, *p* = 0.024) [31].

According to Poole et al. (2010) [32] and Yuen et al. (2011) [33], following a special domestic hygiene regimen had a favorable effect on the clinical parameters “bleeding-on-probing (BOP)” and “gingival index (GI)”. Six months after the beginning of the study, BOP was significantly reduced (2.5 ± 3.7, *p* < 0.05) in those locations where BOP (8.5 ± 21.1, *p* < 0.05) was elicited during the baseline assessment. The GI was reduced by a statistically significant degree (20.8%) in a study population (N = 48) that was specifically instructed in the use of an electric oscillating-head toothbrush and the Reach^®^AccessTM Flosser (a tooth floss that can be used with one hand) [33].

Lo Giudice et al. [35] investigated whether a lower pain threshold is associated with increased temporomandibular dysfunction in SSc in comparison with psoriasis arthritis (PsA) and healthy controls. Based on the Helkimo score, which is an index for jaw mobility [38], pain during movement and palpations were found to be significantly more severe than those in the control group, and the functionality of the masticatory apparatus was compromised.

### 3.2. Additional Analysis—Quantitative Synthesis of Studies

Data from 12 publications (766 patients) were used to compute the pooled effect estimate from which prevalences of oral and/or perioral symptoms in SSc could be derived. Three studies [26,27,30] looked at the relationship between SSc and changes of the oral mucosa. Six studies in total [24,25,29,31,33,34] reported prevalences of the development of labial changes, whereas four publications reported salivary gland changes in [26,27,30,33] the other regions group.

Given the dearth of data from prospective studies, only the main locations “oral mucosa”, “lip”, and “other regions” could be used and evaluated to compute prevalences. The six study reports that produced data relating to the location “lip” furnished a pooled prevalence estimate of 63.1% with a 95% confidence interval of 48.9–75.3%. Patient numbers were between 9 [24] and 200 [25]. The per-study prevalences were between 42.4% [31] and 97.6% [34] (Figure 4).

For the location “oral mucosa”, a pooled prevalence estimate of 35.5% (95% CI: 15.7–62.0%) was computed based on three studies. The number of patients was between 44 [27] and 133 [30], and the per-study prevalence showed values between 20.5% [27] and 58.6% [30] (Figure 5).

The pooled prevalence estimate for the group “other regions”, which only reported salivary gland changes, was determined to be 25.4% (95% CI: 14.2–41.3%) (Figure 6). Sample size was between 44 [27] and 133 [30] participants. The individual studies determined prevalence values between 2.1% [33] and 37.6% [30].

Table 5 gives an overview of the number of studies included, together with patient numbers and pooled effect estimates. Because of the high heterogeneity for I2 = 64.398% (“lip”), 91.958% (“oral mucosa”), and 78.905% (“other regions”), the computed pooled effect estimates were adjusted using Duval and Tweedie’s Trim and Fill [39].

## 4. Discussion

This literature search addressed the problem of the prevalence of oral mucosa changes in scleroderma patients. Whereas the literature contains a multitude of pertinent case reports, publications that detail investigations with a common research question and high level of evidence (RCTs) that could be further evaluated are lacking.

In order to be able to evaluate the prevalence for the present pool of scleroderma studies as objectively as possible, methods used in the preparation of meta-analyses were applied in this analysis. In this manner, the per-study prevalences were pooled by weighting. This weighted prevalence is called the pooled effect estimate, although the magnitude up to which the heterogeneity between individual studies is negligible and their computation is reasonable is not clearly defined [40]. Another point for consideration is the number of included studies in the various study groups, a feature that makes a general statement difficult. No studies for the main groups “bone”, “tongue”, and “periodontal status” were available that met the inclusion criteria. Whereas six publications were evaluated in the group “lip”, only three and four publications for the groups “oral mucosa” and “other”, respectively, could be drawn upon to compute the pooled effect estimate. Another critical aspect is that the number of cases within the studies included must be classified as low and inhomogeneous. Moreover, the publications differ considerably from this work’s subject matter in terms of their aims. This is the reason that an adjusted effect estimate has been computed in the group “lip” by applying Duval and Tweedie’s Trim and Fill method [41].

To the best of our knowledge, this is the first systematic literature review to examine the prevalence of orofacial manifestation involvement in the context of systemic scleroderma using clinical prospective studies. There is no data from other studies in the literature that can be compared with this one. The statistical methodology we selected was used for the first time in this context. Nevertheless, frequencies associated with the results we obtained are reported. On the basis of the above, we need to bear in mind in this context that some systemic scleroderma symptoms might precede a proper diagnosis by several years. In addition to Raynaud’s symptomatology, which has a prevalence of about 90% in female sufferers in relation to systemic scleroderma [42], the detection of orofacial changes, which are typical of scleroderma, also plays a critical role. Albilia et al. [19] confirmed in their review the known extraoral and intraoral changes and concluded that the initial diagnosis of scleroderma can be made by specialists in the field of dental, oral, and maxillofacial surgery.

In about 80% of cases, the orofacial region is the most affected by SSc [43]. Other systemic symptoms of oral problems, as well as dental (decayed, missing teeth), periodontal, and orofacial anomalies (e.g., xerostomia, mandibular bone resorption, or microstomia) [44] can overshadow them. Other dermal manifestations include telangiectasia, frequently in the region around the mouth, cheeks, lips, and nose.

### 4.1. Systemic Sclerosis and Lips

Cutaneous involvement of the orofacial area results in a narrowing of the oral aperture, with circumoral furrows or perioral whistle lines appearing in around 70% of patients, as well as a wide-eyed appearance caused by periorbital fibrosis [19]. Microstomia makes oral hygiene, dental care, and prosthetic rehabilitation more complicated as well as causing mastication and deglutition problems [45]. Voice, facial expressions, and saliva control can be hampered by thin lips and facial power [46]. The development of circumoral furrows (80%), tighter mouth (77%), thin lips (73%) and lack of facial lines (68%) were all found to be significantly more frequent than non-facial SSc appearances in a broad patient survey examining features of SSc that trigger the most cosmetic concern [47].

### 4.2. Systemic Sclerosis and Salivary Glands

Oral health, speech, and swallowing are all affected by salivary hypofunction caused by fibrosis or medications [19,45]. According to Crincoli et al. [48], 78.8% of their study group with SSc complained of oral symptoms like xerostomia compared to controls (28.7%) (χ^2^ = 40.23 *p* = 0.001). In a study of Couderc et al. [49], changes of the salivary glands, i.e., xerostomia, were reported in 40% (N = 10) of the patients with SSc. Actually, sicca symptoms are frequently reported by patients with SSc (7.5–68%) and could be due to a fibrosis process of the salivary glands [30,50,51].

Approximately two-thirds of SSc patients have xerostomia, with the effect ranging from mild to extreme in more than half of these patients [50,52,53]. Xerostomia can be caused by two different mechanisms, according to studies involving minor salivary gland biopsies: immune-mediated destruction of the acinar tissues (as seen in Sjögren’s syndrome) or fibrosis of the salivary glands, which reduces their exocrine ability [50,54,55]. The presence of Sjögren’s syndrome in SSc patients seems to be more closely linked to the lcSSc phenotype [30,50].

Baron et al. (2015) studied a total of 163 SSc patients, of whom 72% (N = 117) and 28% (N = 46) had the limited and diffused types, respectively. Reduced salivation was associated with Sjögren-specific autoantibodies (β = −43.32; 95% confidence interval [95% CI]: −80.89, −5.75) but was not correlated with disease severity (β = −2.51; 95% CI: −8.75, 3.73). On the other hand, a diminished intercisal distance was linked to the severity (β = −1.02; 95% CI: −1.63, −0.42) and the modified Rodnan skin thickness score (β = −0.38; 95% CI: −0.53, −0.23). Reduced salivation was significantly associated with the number of missing teeth (relative risk [RR] 0.97; 95% CI: 0.94, 0.99), impaired manual agility (RR 1.52; 95% CI: 1.13, 2.02), and gastroesophageal reflux disease (GERD; RR 1.68 [95% CI: 1.14, 2.46]) [54].

### 4.3. Systemic Sclerosis and Oral Mucosa

In SSc, poor nutritional intake and vitamin deficiencies can cause oral mucosal atrophy and ulceration [45]. Oral ulceration can also be a side effect of SSc pharmacological treatment, which includes methotrexate, azathioprine, and cyclophosphamide [56]. Correction of any nutritional deficiencies, topical treatment of the ulcers, such as covering agents, and substitution of immunosuppressant medication as required will all be part of the treatment plan. Persistent, non-healing ulcers should be treated with caution, and a biopsy may be necessary to rule out more serious pathology.

According to the literature, a high concentration of squamous cell carcinoma occurs in the oral mucosa of SSc patients [20,57,58]. For this reason, every possibility should be explored to perform an exhaustive oral assessment of SSc patients, even when the mouth opening is impaired (e.g., by using an endoscope) [22].

Jackowski et al. (2002) described orofacial changes in scleroderma within four categories: physiognomy, lips, tongue, and oral mucosa. Their 70 patients (mean age 54.4 ± 11.4 years, S.D.) with scleroderma underwent a clinical orofacial examination; 70 age- and sex-matched patients (mean age 49.8 ± 15.9 years, S.D.) with no history of rheumatic disease were selected at random as a control group. Thirty-five orofacial parameters were evaluated by one single calibrated investigator and classified into three groups (“normal”, “changed”, and “severely changed”). Analysis of the check-list indicated that the distribution of the groups “normal”, “changed”, and “severely changed” varied significantly within the four categories (physiognomy, lips, tongue, and oral mucosa) (*p* < 0.01). On the basis of the homogeneous distribution pattern over the three degrees of severity, both the clinical examination of physiognomy and, to a limited degree, the lips and the tongue can be used to support the diagnosis of scleroderma. Whereas the clinical examination of the oral mucosa showed a disproportionate distribution of the patients in the first group (“normal”), 11 of the 35 examination criteria investigated in this study can be used to support the diagnosis of scleroderma from a dentist’s point of view [59].

## 5. Conclusions

A distinct paucity exists with regard to studies of a prospective character looking into the oral symptoms of SSc. The present systematic literature review shows a concentration of orofacial changes in SSc sufferers. The oral cavity is involved in the pathological course of SSc to variable degrees. The most pathognomonic conditions in the orofacial region in patients with systemic scleroderma affect the lips with a prevalence of 57.6% (95% CI: 40.8–72.9%), oral mucosa with a prevalence of 35.5% (95% CI: 15.7–62.0%), and salivary glands with a prevalence of 25.4% (95% CI: 14.2–41.3%).

Further studies are necessary to determine a more precise prevalence of oral/perioral changes typical of SSc and of symptoms in the oral and maxillofacial area. Supporting diagnostic biopsies for oral markers has not yet been an option in the clinical management of SSc patients; therefore, an evaluation of orofacial changes by visual diagnostic assessments is of critical importance.

### Limitations

The aim of this analysis was to summarize the results of existing studies in the field of indication of scleroderma. Due to the fact that no information regarding the prevalence of scleroderma in special areas of the oral cavity exists, we tried to evaluate a pooled effect estimate. Because there is only low evidence by study design (RCT) in publications of the research area, and prevalences were extracted from studies with different study primary endpoints, we used meta-analysis methods and present the results as pooled effect estimates. In order to obtain the most valid information possible (clinical trials of high quality), this analysis comprises only prospective studies with a minimum of ≥3 patients. This approach was defined as part of the evaluation and might cause distortions due to changes in the study size or study design. However, this approach was preferred for the highest validity possible.

Another limitation of our analysis arises from the substantial heterogeneity in the reported data, as considerable heterogeneity was ascertained. The reader should bear this in mind while interpreting the results.

Furthermore, bias cannot be ruled out, owing to the procedure of data collection used in this project.

The study quality in the analyzed publications shall also be addressed here. Only 14 out of 193 reports were found to be eligible for this review (with a prospective study design). From these publications, only six documented ethical approval (43%), and three publications stated a statistical department for planning or analysis. In two studies, a flow chart was available, and only one trial reported a sample size calculation/sample size legitimation. In total, the area of scleroderma shows low validity in terms of study design, so a reliable and valid calculation of prevalence is difficult.

## Figures and Tables

**Figure 1 ijerph-18-05238-f001:**
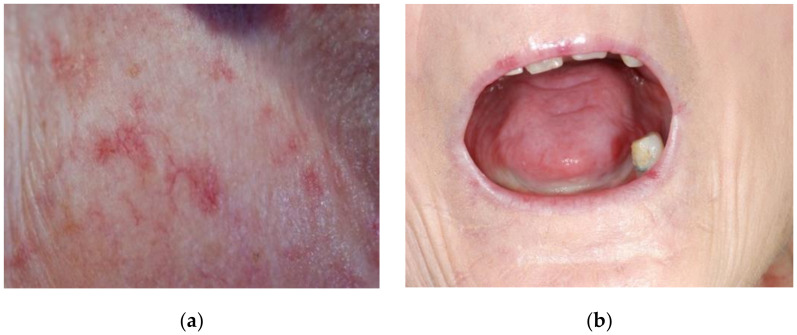
Typical orofacial manifestations of patients suffering from SSc: (**a**) perioral telangiectasia caudal of the right infraorbital margin; (**b**) Xerostomia, tongue smooth and atrophic.

**Figure 2 ijerph-18-05238-f002:**
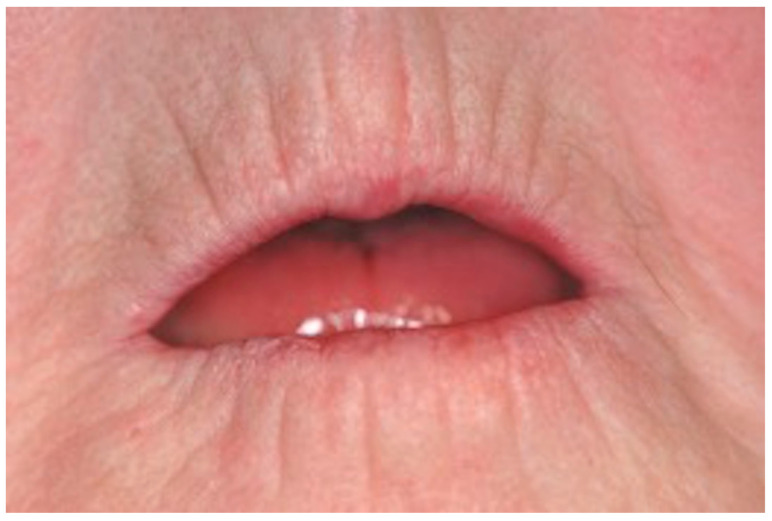
Patient suffering from SSc. Perioral wrinkles, microcheilia, microstomy.

**Figure 3 ijerph-18-05238-f003:**
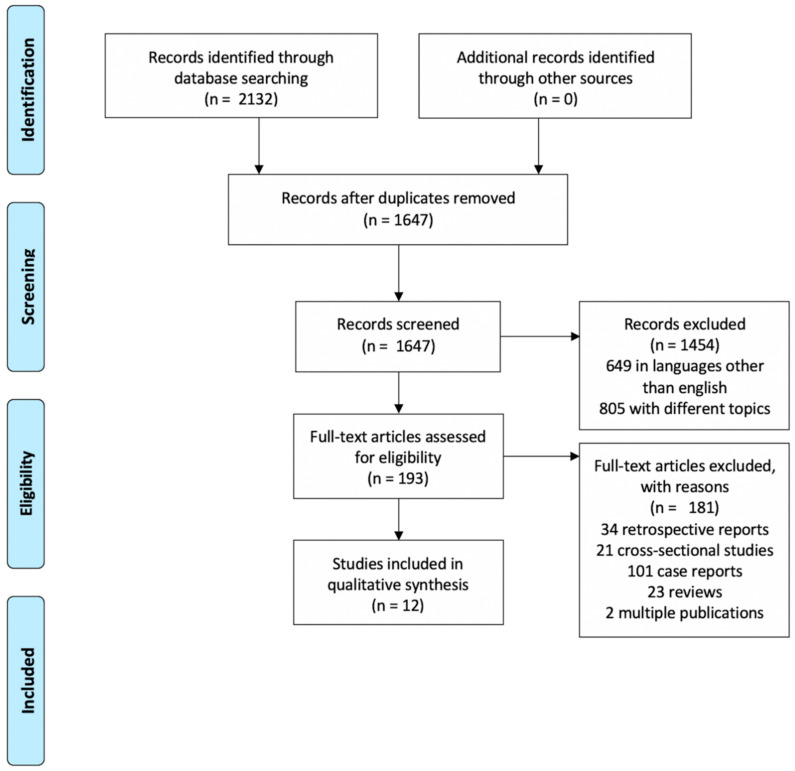
Flow chart for selection of records.

**Figure 4 ijerph-18-05238-f004:**
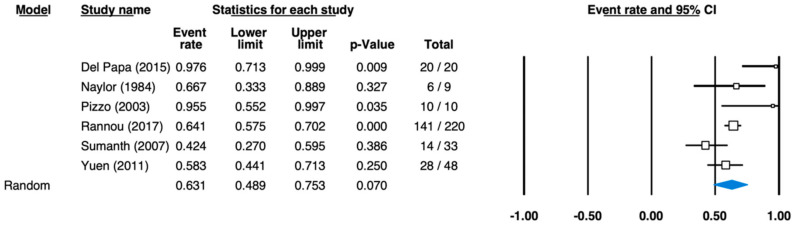
Forest plot for the pooled effect estimates of prevalence in scleroderma—lip symptoms (heterogeneity: I-squared = 78.905; tau-squared = 0.375, *p* = 0.003).

**Figure 5 ijerph-18-05238-f005:**
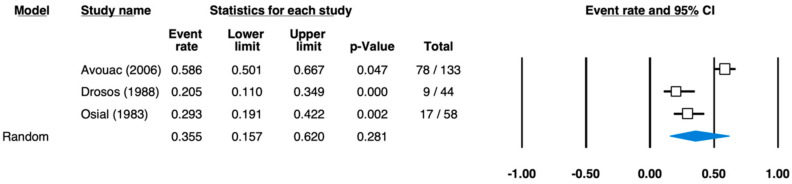
Forest plot for the pooled effect estimates of prevalence in scleroderma—oral mucosa symptoms (heterogeneity: I-squared = 91.958; tau-squared = 0.835, *p* = 0.000).

**Figure 6 ijerph-18-05238-f006:**
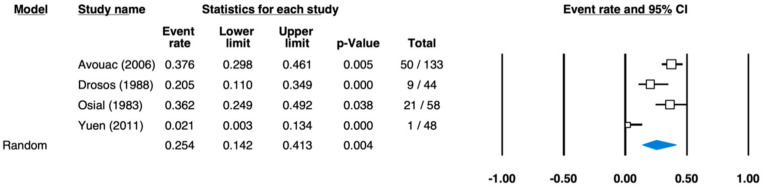
Forest plot for the pooled effect estimates of prevalence in scleroderma—other symptoms (heterogeneity: I-squared = 78.905; tau-squared = 0.375, *p* = 0.003).

**Table 1 ijerph-18-05238-t001:** Inclusion and exclusion criteria.

Inclusion Criteria
Language: English Study design: prospective clinical trial and case series with at least 3 patients Number of patients: ≥3 Therapy: no restrictions regarding the therapy
**Exclusion Criteria**
Reviews, meta-analysis Retrospective study design or no randomization evident Animal experiments Laboratory studies ”Questionnaire”, published ”study protocol” ”Letter to the editor”, ”commentary”, ”viewpoint” Device evaluation

**Table 2 ijerph-18-05238-t002:** Search algorithm used.

**Search algorithm**	(“systemic sclerosis” OR “scleroderma”) AND (“oral” OR “mouth” OR “gingiva” OR “intraoral” OR “perioral” OR “tongue” OR “alveolar” OR “periodontium” OR “periodontal” OR “jaw” OR “jaws” OR “mandible” OR “maxilla” OR “gnathic” OR “facial” OR “craniofacial” OR “maxillofacial” OR “temporomandibular joint” OR “palate” OR “palatal” OR “palatine” OR “palatum” OR “palatinal” OR “palatopharyngeal” OR “zygomatic” OR “uvula” OR “salivary” OR “parotid” OR “sublingual” OR “Sjogren’s syndrome” OR “Sjogren syndrome” OR “oral hygiene”) AND (“treatment” OR “therapy”)

**Table 3 ijerph-18-05238-t003:** Search terms divided in main groups and subgroups.

Main Group	Subgroups	Main Group	Subgroups
Oral Tissue	Oral mucosa	Tongue	Mobility
	Palatinal rugae		Plasticity
	Palatopharyngeal arch		Swallowing disorders
	Hard palate		Consistence
	Soft palate		Colour
	Uvula		Atrophy
	Base of the mouth		Raynaud-phenomenon
	Gingiva		Fibrosis
	Intraoral telangiectasias		Lingual frenulum/scleroglosson
	Xerostomia		
	Capillary system		
	Atrophy		
Lips	Microstomia	Periodontal status	Periodontal ligament
	Microcheilia		Periodontitis
	Colour		
	Consistence		
	Ulcerations		
	Mouth angle		
	Perioral wrinkles		
	Capillary system		
	Labial frenulum		
Bone	Alveolar bone	Other	Muscles
	Jaw bone		Oral cancer
	Ascending ramus		Teeth
	Jaw angle		Trigeminal nerve
	Zygomatic arch		Salivary glands
	TMJ and coronoid process		
	Dysgnathic alterations		

**Table 4 ijerph-18-05238-t004:** Studies included in the evaluation.

Author and Year	Cases	Age	Sex	Disease Duration (Years)	Methods	Results
Osial et al., 1983 [26]	58	51 ± 2	F: 86% M: 14%	6.1 ± 1.0	Clinical examination of the patients and histopathological assessment of the labial small salivary glands to investigate Sjögren’s syndrome	Enlargement of the labial salivary glands (2), fibrosis of the salivary glands (19), Sjögren’s syndrome (17), enlargement of the parotid salivary gland
Naylor et al., 1984 [24]	9	NA	NA	NA	Non-surgical improvement of the mouth opening by forming two groups and comparing two exercises by applying a double-blind procedure	Mean mouth opening improvement in control group: 3.0 mm; mean mouth opening improvement in test group: 5.6 mm
Drosos et al., 1988 [27]	44	49.2 ± 13.3	F: 95% M: 5%	8.0 ± 7.4	Evaluation of simultaneous existence of Sjögren’s syndrome via lip biopsy, dry keratoconjunctivitis, and/or xerostomia	Labial salivary gland score: 2 + (10), 1 + (3); mild to moderate fibrosis (17); normal tissue (14); Sjögren’s syndrome (9); enlargement of the parotid gland (20);
Clegg et al., 1994 [28]	146	49 ± 13	F: 83% M: 17%	104 months	Comparison between potassium aminobenzoate and a placebo in the treatment of the dermal manifestations	No significant improvement as to mouth opening, skin thickness, and lip mobility through medication
Pizzo et al., 2003 [29]	10	56.8 ± 11.19	F: 100%	NA	Investigation of the effect of an 18-month, non-surgical, domestic-exercise-based intervention on mouth opening	All subjects showed improved mouth opening
Avouac et al., 2006 [30]	133	55 ± 13	F: 86% M: 14%	6.5 ± 6	Subjective mouth dryness questionnaire, Schirmer I test for measuring the salivary flow rate; labial biopsy after positive questionnaire or Schirmer test	Subjective sicca symptomatology (85); positive Schirmer I test (61); biopsy (91)
Sumanth et al., 2007 [31]	33	31 ± 9	F: 88% M: 12%	5.6 ± 4.5	Effects of methotrexate administration on oral mouth health	Statistically significant improvement of mouth opening (33.92 mm ± 1.70 mm, *p* = 0.024) (25); no statistical differences in histopathological assessment
Poole et al., 2010 [32]	17	53.9	F: 89% M: 11%	10.75	Effect of oral hygiene measures and domestic exercises on oral mouth health	Significant reduction of oral inflammation parameters (BOP) after 6 study months
Yuen et al., 2011 [33]	48	50.7 ± 13.0	F: 79% M: 21%	7.6 ± 6.1	Effect of an oral hygiene regime on gum health	Significant improvement of gingival index (GI)
Del Papa et al., 2015 [34]	20	35 ± 15	F: 100%	11 ± 10	Effect of autologous fat transplantation to treat perioral fibrosis	Significant improvement of mouth opening and function
Rannou et al., 2017 [25]	218	52.7 ± 14.8 (Group 1) 53.1 ± 14.4 (Group 2)	F: 86 % M: 14% (Group 1) F: 80% M: 20% (Group 2)	6.5 ± 6.5 (Group 1) 6.7 ± 8.6 (Group 2)	Group 1: Customized physiotherapy in addition to regular therapy; Group 2: No customized physiotherapy in addition to regular therapy	No significant improvement through additional measures after 12 months
Lo Giudice et al., 2018 [35]	30	60	F: 100%	10	To investigate whether a lower pain threshold is associated with increased temporomandibular dysfunction in systemic sclerosis (SSc) compared with psoriasis arthritis (PsA) and healthy controls	The temporomandibular apparatus is functionally impaired in comparison with control group

NA = not available.

**Table 5 ijerph-18-05238-t005:** Overview of the pooled effect estimates of prevalence.

Group	N Studies	N Included Patients	Pooled Prevalence Estimate	N Added Studies	Adjusted Pooled Prevalence Estimate
Lip	6	340	63.1% [95% CI: 48.9–75.3]	2	57.7% [95% CI: 40.8–72.09]
Oral mucosa	3	235	35.5% [95% CI: 15.7–62.0]	0	35.5% [95% CI: 15.7–62.0]
Other	4	283	25.4% [95% CI: 14.2–41.3]	0	25.4% [95% CI: 14.2–41.3]

CI = Confidence interval.

## Data Availability

Not applicable.

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
