# Peer review of "Prevalence of Oral and Maxillofacial Disorders in Patients with Systemic Scleroderma—A Systematic Review"

_ijerph, 2021, doi:10.3390/ijerph18105238_

Round 1
Reviewer 1 Report
No any other comments
Author Response
Reviewer 1:
No any other comments
Answer:
Thank you very much.
Reviewer 2 Report
This manuscript is a review of systemic scleroderma with a focusing oral manifestation. Authors demonstrated that the lips were mostly affect with a prevalence of 57.6% followed by oral mucosa and salivary gland with a 35.5% and 25.4% prevalence, respectively. These results are interesting but, unfortunately, the discussion is poorly described. Therefore, I think that major revision has to be necessary for acceptance.
Queries and recommendation
Major points
- Discussion should be written based on the results. Please further discuss your results.
Minor points
- “3.1. Results of individual studies”:
- Five paragraphs are documented but the subjects are the numbers of selected articles, Sjörgen's syndrome, oral exercise, medication, oral hygiene, temporomandibular function. The 4th paragraph has to be divided into two paragraphs of medication and oral hygiene after the line 241 from “According to Pooles et al.”
- Some sentences are confused with the other subjects. The line 220-223 and 234-236 have to move to the next paragraph.
- Table 1: The second column should be aligned to the left.
- On the line161, please add state name.
Author Response
Reviewer 2:
This manuscript is a review of systemic scleroderma with a focusing oral manifestation. Authors demonstrated that the lips were mostly affect with a prevalence of 57.6% followed by oral mucosa and salivary gland with a 35.5% and 25.4% prevalence, respectively. These results are interesting but, unfortunately, the discussion is poorly described. Therefore, I think that major revision has to be necessary for acceptance.
Queries and recommendation
Major points
- Discussion should be written based on the results. Please further discuss your results.
Minor points
- “3.1. Results of individual studies”:
- Five paragraphs are documented but the subjects are the numbers of selected articles, Sjörgen's syndrome, oral exercise, medication, oral hygiene, temporomandibular function. The 4th paragraph has to be divided into two paragraphs of medication and oral hygiene after the line 241 from “According to Pooles et al.”
- Some sentences are confused with the other subjects. The line 220-223 and 234-236 have to move to the next paragraph.
- Table 1: The second column should be aligned to the left.
- On the line161, please add state name.
Anwer:
Thank you very much for your answer!
Major points
Thank you very much for this point! We rewrote the discussion section and focused on our results by paragraphing them into subheadings. We hope this helps to clarify this part of the manuscript.
Minor points
The 4th paragraph ist now divided into two paragraphs which makes absolutely sense, thank you very much for this point!
We moved the lines 220-223 and 234-236 to the next paragraph.
Table 1: The second column is aligned to the left now.
The state name in line 161 has been added.

Reviewer 3 Report
Dear authors,
it is a very interesting work. Only two questions:
1) in the abstract you said "up to November 2019", but in the review you have included works published also in 2020. Please clarify this point.
2) previous works with 3 patients are case series and not trial, so it is better to add this aspect in the inclusion criteria (clinical trial and case series with at least 3 patients). Please improve it.
In my opinion could be very useful for readers to insert some Images of the most important scleroderma lesions on lips and oral cavity.
Congratulations
Kind regards
Author Response
Reviewer 3:
Dear authors,
it is a very interesting work. Only two questions:
1) in the abstract you said "up to November 2019", but in the review you have included works published also in 2020. Please clarify this point.
2) previous works with 3 patients are case series and not trial, so it is better to add this aspect in the inclusion criteria (clinical trial and case series with at least 3 patients). Please improve it.
In my opinion could be very useful for readers to insert some Images of the most important scleroderma lesions on lips and oral cavity.
Congratulations
Kind regards
Answer:
Thank you very much for your answer!
1, The abstract was changed so that the final inclusion date was March 2020. This was a mistake on our part, thank you very much for this important information!
2, We improved the inclusion criteria by this important note, thank you very much for this point!
We have added a few clinical images to the introduction part to help the reader understand the clinical phenotypes better.
Round 2
Reviewer 2 Report
This manuscript is a well-written paper of systemic scleroderma with a focusing oral manifestation. Re-submitted manuscript has been mostly revised except for the citations style for MDPI. Please confirm instructions for authors. You should either cite all authors, or cite the first ten authors, and when more than ten add ‘et al.’ at the end.
This manuscript is a resubmission of an earlier submission. The following is a list of the peer review reports and author responses from that submission.
Round 1
Reviewer 1 Report
Answer: The challenge in the search was that very few studies met the criteria of a systematic review. For this reason, all studies were included in the analysis that had a prospective design and reported orofacial involvement, regardless of the therapy being investigated. For the individual questions (scleroderma and temporomandibular joint treatment or scleroderma and treatment of oral and maxillofacial malformations or scleroderma and Sjögren's syndrome, etc.), there is no sufficient data available according to our search strategy. Table 4 lists the publications that provided evaluable data using proven statistical methods. Therefore, this table must also remain listed in order to be able to present the methodology in a comprehensible way.
However, the studies which the researchers have used, do not answer the aim: which orofacial symptoms are most common. If there is not enough published data then, perhaps, the meta-analysis cannot be yet performed. I believe authors need to change their aims.
Author Response
However, the studies which the researchers have used, do not answer the aim: which orofacial symptoms are most common. If there is not enough published data then, perhaps, the meta-analysis cannot be yet performed. I believe authors need to change their aims.
Answer:
Thank you very much for your answer and your suggestion! As formulated in the "Discussion" section, our main research question is not at the same time the primary research question in the underlying study reports, as is required for conducting meta-analyses.
The aim was not to find out the effects of specific therapies or results of studies in this patient clientele, but to filter out and evaluate all described orofacial changes as "secondary findings", as to our knowledge there have been no RCTs in connection with systemic scleroderma and orofacial changes so far.
In this project, prevalences were calculated on the basis of the available study reports and the case numbers contained therein (total case number and case number of the reported incidences of the disease in the different structures of the oral cavity). The aim was not to examine the incidences, i.e. newly occurring changes in the orofacial area in connection with a specific therapy, but the prevalences, i.e. the actual observation of orofacial changes - independent of the therapy or examination initiated in each case.
Our study shows that there are scientifically evaluable data. However, we apologise for our misleading wording; we have now modified and clarified our objective (see lines 108-111).
Reviewer 2 Report
In looking over the PICO format and examining the various Scleroderma symptomologies, Retrospective data was excluded because it doesn't fit with regard to Cochrane meta analysis concept. I believe that the Cochrane process is not the best way to evaluate for prevalence. How many of the included studies were within institutions with previous retrospective data? I would like to see such. I believe that prevalence studies are more accurate when taken from public health registries. Looking at RCTs and especially RCTs that are heterogeneous is a problem.
The authors essentially noted the above in the limitations of the study and should be applauded for noting such limitations. But I believe they should have known about these limitations before selecting this mode of data gathering. I think the title should note incidence rather than prevalence and the references to prevalence should be amended to incidence/prevalence and a paragraph included to address this issue.
Author Response
In looking over the PICO format and examining the various Scleroderma symptomologies, Retrospective data was excluded because it doesn't fit with regard to Cochrane meta analysis concept. I believe that the Cochrane process is not the best way to evaluate for prevalence. How many of the included studies were within institutions with previous retrospective data? I would like to see such. I believe that prevalence studies are more accurate when taken from public health registries. Looking at RCTs and especially RCTs that are heterogeneous is a problem.
The authors essentially noted the above in the limitations of the study and should be applauded for noting such limitations. But I believe they should have known about these limitations before selecting this mode of data gathering. I think the title should note incidence rather than prevalence and the references to prevalence should be amended to incidence/prevalence and a paragraph included to address this issue.
Answer:
In this project, prevalences were calculated on the basis of the available study reports and the case numbers contained therein (total case number and case number of the reported occurrence frequencies of the disease in the different structures of the oral cavity). The aim was not to examine the incidences, i.e. newly occurring changes in the orofacial area in connection with a specific therapy, but the prevalences, i.e. the actual observation of orofacial changes - independent of the therapy or examination initiated in each case. However, we apologise for our misleading wording; we have now modified and clarified our objective (see lines 108-111).
According to the flow diagram, 34 publications were excluded from a pool of 193 matching by topic and language due to retrospective study design.
More than 50% of the data basis shows the lowest evidence -> case reports (101 publications).
This distribution already shows that for a summary assessment (i.e. for the presentation of the previous study results in the form of a meta-analysis or pooled effect estimate) it was necessary to fall back on lower evidence.
However, it also turned out that there were 12 prospective studies that were then selected on the basis of the - by definition - available higher evidence.
Of course, the pool for evaluation could have been increased in number by combining prospective, retrospective and cross-sectional studies, but this does not correspond to the procedures of meta-analyses. We were aware of this and for this reason decided to prepare a review with additional calculation of a pooled effect estimate. In addition, if retrospective data had been used, the correct and comprehensive data from the patient files could have been justifiably questioned, or in other words: how much data could have been included here on the basis of the inspection of the files, and would this not also have led to a bias in the data basis?
The heterogeneity is rather due to the very open literature search. If the search had been restricted to a specific indication or therapy, the data pool would have been greatly reduced. In the "manual selection" we checked the indication-related inclusion criteria in order to generate the largest possible amount of data that was necessary for the research question.
Reviewer 3 Report
No more comments.
Author Response
Thank you very much!